# CURRICULUM-AWARE TRAINING FOR DISCRIMINATING MOLECULAR PROPERTY PREDICTION MODELS

**Hansi Yang**
Department of Computer Science and Engineering
Hong Kong University of Science and Technology
Hong Kong, China
`hyangbw@cse.ust.hk`

**Quanming Yao**
Department of Electronic Engineering,
State Key Laboratory of Space Network
and Communications, Tsinghua University
Beijing, China
`qyaoaa@tsinghua.edu.cn`

**James Kwok**
Department of Computer Science and Engineering
Hong Kong University of Science and Technology
Hong Kong, China
`jamesk@cse.ust.hk`

## ABSTRACT

Molecular property prediction plays a crucial role in various fields such as cheminformatics and artificial intelligence. Despite its wide applicability, current models still struggle in the presence of activity cliff, in which molecules with similar chemical structures display remarkable different properties. This hinders the model's ability to learn distinctive representations for molecules with similar chemical structures, resulting in inaccurate predictions on molecules with activity cliff. In this paper, we first present empirical evidence demonstrating the ineffectiveness of standard training pipelines on these molecules. We then propose a novel approach that reformulates molecular property prediction as a node classification problem, and introduce both node-level and edge-level tasks to improve the learning for these challenging molecules. The proposed method is versatile, and can be seamlessly integrated into a variety of pre-trained or randomly initialized base models. Extensive evaluation on various molecular property prediction datasets validate the effectiveness of our approach.

## 1 INTRODUCTION

Molecular property prediction aims to determine the properties of molecules directly from their chemical structures. It plays a crucial role in various fields, including drug discovery (Stokes et al., 2020), material science (Chanussot et al., 2021; Tran et al., 2023) and bioinformatics (Narayanan et al., 2002; Zhou et al., 2023). Despite its broad applicability, recent studies (van Tilborg et al., 2022; Deng et al., 2023) show that current models often fail to generate sufficiently discriminative molecular representations. sometimes, they can even perform worse than models using fixed representations (e.g., molecular fingerprints). This problem arises as existing machine learning models tend to produce similar representations for structurally similar molecules. When two such molecules exhibit different properties, accurately predicting their properties becomes challenging due to their indistinguishable representations. This phenomenon, referred as activity cliff (AC) (Stumpfe et al., 2019; Tamura et al., 2023; Dablander et al., 2023), is prevalent across various molecular property datasets. Figure 1 shows an example from the *Tox21* data set (Wu et al., 2018). Here, the two molecules have only

(a) Retinol.

(b) Retinal.

Figure 1: Examples of two molecules with AC

minor differences (the two yellow boxes), but their responses to the ER, ATAD5 and HSE receptors are all different.

While numerous studies (Maggiora, 2006; van Tilborg et al., 2022; Graff et al., 2023; Deng et al., 2023) have verified that AC causes difficulties for existing molecular property prediction models, their analysis focus only on the *inference* stage. It remains unclear why these models fail to learn discriminating molecular representation during *training*. Similar to inference, training a model to differentiate structurally similar molecules with distinct properties inherently presents challenges. Nevertheless, no existing work considers how to address this challenge. Standard training pipelines only lead to models that are incapable of distinguishing molecules with AC.

Motivated by the shortcoming of existing training algorithms in obtaining discriminative molecular representations, in this paper, we propose a novel training algorithm to enhance learning from molecules with AC. Through empirical analysis, we first demonstrate that standard training algorithms struggle to accurately fit molecules with AC during training, and this challenge persists across different model backbones and pre-training tasks. To alleviate this problem, we propose a new training algorithm that focuses on improving the model's discriminative power by effectively learning from molecules with AC. We first reformulate molecular property prediction as a node classification problem on graphs, where each node represents a molecule, and edges are defined by similarities in their chemical structures. We then introduce two tasks at the node and edge levels respectively. For the node-level task, we employ curriculum learning that considers both the loss and AC information in the selection of informative molecules for model training. For the edge-level task, we introduce a novel pairwise modeling task to align the model directly with AC on different molecular properties. The proposed method can be integrated with different base models, pre-trained . or randomly initialized. Empirical results on various molecular property prediction data sets demonstrate effectiveness of the proposed method.

Our contributions can be summarized as follows:

- We are the first to investigate why existing molecular property prediction models fail to produce discriminative molecular representations. Using molecules with AC as representatives, we show that standard training pipelines struggle to accurately fit these molecules, a limitation observed in both randomly-initialized and pre-trained models.
- We propose to reformulate molecular property prediction as a node classification problem. We then introduce two novel tasks at the node and edge levels, so as to learn from molecules with AC more effectively and produce models with good discriminative ability.
- Empirical results on various molecular property data sets demonstrate that the proposed method improves the performance of both random-initialized and pre-trained models.

## 2 RELATED WORKS

### 2.1 MOLECULAR PROPERTY PREDICTION WITH GRAPH NEURAL NETWORKS

Molecular property prediction predicts the molecular properties from a molecular graph, in which each node is an atom and each edge is a chemical bond between atoms. Naturally, various graph learning architectures can be applied. Pioneering works (Merkwirth & Lengauer, 2005; Gilmer et al., 2017) use the message-passing graph neural networks (GNN) (Veličković et al., 2018; Xu et al., 2019). However, the GNN may not be able to capture long-range dependencies (Rampášek et al., 2022). Instead, recently, transformer models (Vaswani et al., 2017) are used to model long-range interactions between nodes (Ying et al., 2021; Rampášek et al., 2022). For example, EGT (Hussain et al., 2022) uses global self-attention to update both the node and edge representations for quantum-chemical regression. This allows unconstrained dynamic long-range interactions between nodes, and results in better performance.

Besides using various deep learning architectures, another approach improves performance by using different graph pre-training tasks. Most of these works consider how to effectively use the geometric information contained in the 3D conformers of different molecules (Townshend et al., 2019; Axelrod & Gomez-Bombarelli, 2022). For example, Klicpera et al. (2020) uses the relative 3D information (such as bond length and bond angle) derived from the absolute Cartesian coordinates. GemNet (Gasteiger et al., 2021) further captures information from the dihedral angle to uniquely define all

relative atom positions. SphereNet (Liu et al., 2021) proposes a generic framework for the 3D graph network, and designs a spherical message passing mechanism. 3D Infomax (Stärk et al., 2022) proposes to maximize mutual information between the 3D structures and representations from the GNN, enabling the model to produce implicit 3D information that can be useful for the downstream tasks. 3D-PGT (Wang et al., 2023b) proposes a multi-task 3D pre-training framework that predicts bond length, bond angle and dihedral angle from molecular graphs. UniMol (Zhou et al., 2023) proposes to jointly use the 3D position recovery task and masked atom prediction task for pre-training, and achieves state-of-the-art performance on various molecular property prediction benchmarks.

The negative impacts of AC to molecular property prediction have long been investigated (Maggiora, 2006; van Tilborg et al., 2022; Graff et al., 2023; Deng et al., 2023). However, they focus on the inference stage, while we propose to confront its negative impacts with a novel training algorithm. Some other works (Horvath et al., 2016; Iqbal et al., 2021; Park et al., 2022; Zhang et al., 2023; Wu, 2024) predict whether a given pair of molecules have AC, which differs from our focus on solving molecular property prediction.

## 2.2 CURRICULUM LEARNING

Curriculum learning (CL) (Wang et al., 2022) first trains a learning model with easier training samples so that the model can easily obtain a coarse decision boundary. The model is then refined by harder samples later in the training process. As an easy-to-use plug-in, curriculum learning has shown improved generalization performance of various models in a wide range of scenarios, including computer vision (Guo et al., 2018), natural language processing (Platanios et al., 2019; Liu et al., 2020) and reinforcement learning (Narvekar et al., 2017).

Curriculum learning has also been applied to graph learning (Wei et al., 2022; Wang et al., 2023a). CLNode (Wei et al., 2022) proposes to jointly consider the loss and node labels in the curriculum learning of node classification. MotifNet (Wang et al., 2023a) uses curriculum learning for motif-based graph learning, and orders the various motifs based on their difficulty levels. CurrMG (Gu et al., 2022) further considers using curriculum learning in molecular property prediction. Nevertheless, their approach only yields limited improvements as they consider the prediction error and molecular structure for each molecule separately, while the proposed method considers the pairwise relationship between molecules.

## 3 CASE STUDIES ON MOLECULES WITH ACTIVITY CLIFF

To see how existing models suffer from limited abilities in distinguishing molecules with similar chemical structures, we take the set of molecules with AC as an example. Loosely speaking, AC refers to a pair of molecules with similar structures but distinct properties. Its precise definition depends on how we characterize structural similarity. In the following, we build upon the definition of matched molecule pairs (Dablander et al., 2023).

**Definition 3.1** (Matched Molecule Pair: Dablander et al. (2023)). A *matched molecule pair* is a pair of molecules that share a common *structural core* (which contains at least twice as many heavy atoms[1] as in the variable parts) but differ by small *variable parts* (which contains no more than 13 heavy atoms) from the chemical transformation of bond cutting on exocyclic bonds.

The definition of AC can then be given as follows:

**Definition 3.2** (Activity Cliff (AC)). Activity cliff refers to a matched molecule pair with different labels with respect to a given property.

Note that the definition of AC depends on the property being considered. For a pair of molecules with similar chemical structures, it may exhibit activity cliff on one property but not another.

While many works have demonstrated the difficulty of making accurate predictions on molecules with AC (Maggiora, 2006; van Tilborg et al., 2022; Graff et al., 2023; Deng et al., 2023), it remains unclear why such difficulty arises, and why existing models cannot produce discriminating representations on these molecules. To empirically investigate these issues, we consider four tasks from the *Tox21* data

---

[1]Heavy atoms are atoms other than hydrogen.

set (Wu et al., 2018) which predict a molecule's response to different receptors (NR-AhR, NR-ER, SR-ARE and SR-MMP). We use two graph neural network models that have been commonly used for molecular property prediction: (i) GIN (Xu et al., 2019) as a representative for message-passing neural networks, and (ii) GraphGPS (Rampášek et al., 2022) as a representative for attention-based graph learning models. Besides training the GIN or GraphGPS models from scratch, we also include two recent state-of-the-art pre-trained models: 3D-PGT (Wang et al., 2023b) and Uni-Mol (Zhou et al., 2023), both of which use attention-based graph learning models similar to GraphGPS.

Figure 2 shows the proportion of molecules with AC among molecules with the top-$n\%$ training loss values. While only about 40% of all samples in the Tox21 data set have activity cliff (Table 8 in Appendix B), **molecules with AC make up a significantly higher proportion of large-loss molecules** (about 60% in samples with the top-10% loss). In other words, activity cliff is a critical source for samples that are not accurately learnt. This also indicates the inability of current models in distinguishing structural similar molecules.

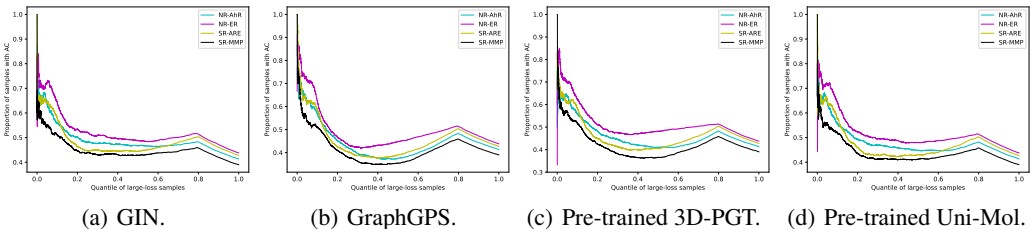

| (a) GIN. | (b) GraphGPS. | (c) Pre-trained 3D-PGT. | (d) Pre-trained Uni-Mol. |

Figure 2: Proportion of molecules with AC among molecules with top-$n\%$ loss values.

Figure 3 shows the training loss curves for the top 10%-loss molecules with and without AC. We can see that **even for these "hard" molecules, molecules with AC have significantly larger training losses** than those without AC. Moreover, even at the end of the training process, the average loss on molecules with AC is still much larger than zero, indicating that the four models do not learn these molecules well with standard training pipelines.

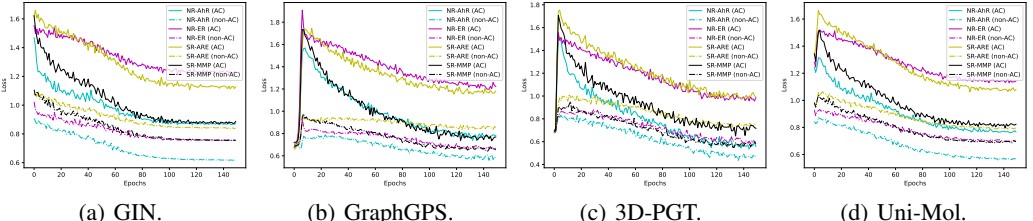

| (a) GIN. | (b) GraphGPS. | (c) 3D-PGT. | (d) Uni-Mol. |

Figure 3: Training losses of large-loss molecules with and without activity cliffs in four model training setups.

In order to accurately distinguish a pair of molecules with activity cliff, the model's decision boundary is expected to separate them even though they have very similar chemical structures. In this experiment, we learn two models, one uses only molecules with AC (denoted "AC") as training samples, while the other uses only molecules without AC (denoted "non-AC"). The randomly-initialized GraphGPS model and pretrained 3D-PGT model are used as base models. Table 1 shows the ROC-AUC values of the two models on various data sets. As expected, using only molecules without AC for training yields worse performance than training on all molecules. Using molecules with AC for training leads to some improvements on the Tox21 and ToxCast data sets. However, the improvements are still limited, as we ignore information from molecules without AC.

Since molecules with AC are both difficult and useful for model training, in the next section, we propose a more effective approach to learn from molecules with activity cliff and obtain a molecular property prediction model with discriminating molecular representation.

## 4 EFFECTIVE LEARNING FROM SAMPLES WITH ACTIVITY CLIFF

Since molecules with AC are both difficult and useful for model training, in this section, we propose a novel training algorithm to effectively learn from molecules with AC for more discriminative

Table 1: ROC-AUC on different molecular property prediction data sets when only using molecules with/without AC for training.

| Method | Tox21 | ToxCast | Sider | MUV | Bace | BBBP | ClinTox | HIV |
|---|---|---|---|---|---|---|---|---|
| GraphGPS (all samples) | 71.5 | 68.5 | 56.4 | 66.9 | 76.9 | **67.0** | **71.1** | **77.0** |
| GraphGPS (AC only) | **71.8** | **69.2** | **56.5** | 68.8 | **77.6** | 67.0 | 67.8 | 72.2 |
| GraphGPS (non-AC only) | 67.8 | 66.9 | 56.3 | **69.2** | 75.8 | 66.3 | 67.4 | 74.8 |
| 3D PGT (all samples) | 73.8 | 69.2 | **60.6** | 69.4 | 80.9 | 72.1 | **79.4** | 69.4 |
| 3D PGT (AC only) | **74.0** | **70.1** | 59.7 | 67.3 | 79.9 | 68.6 | 69.1 | 68.7 |
| 3D PGT (non-AC only) | 68.6 | 68.9 | 58.6 | 64.6 | 79.1 | 65.7 | 77.3 | 69.1 |

molecular representations. We first reformulate molecular property prediction as a node classification problem, with the graph structure induced by the structural similarity in Section 4.1. In Section 4.2, we propose a novel sample selection method that gradually selects hard molecules with AC for training. We further propose a novel edge-level task to align the model with AC on different properties in Section 4.3.

## 4.1 MOLECULAR PROPERTY PREDICTION AS NODE CLASSIFICATION

Given a set of molecules, the definition of matched molecule pairs (Definition 3.1) naturally induces a graph $\mathcal{G} = (\mathcal{V}, \mathcal{E})$. Each molecule corresponds to a node, whose features correpond to the molecular representation obtained by the pre-trained models. Two nodes (molecules) are connected if they are a matched molecule pair. Figure 4 shows an example subgraph for seven molecules on two property prediction tasks (responses to ARE and MMP receptors) from the *Tox21* data set. As mentioned in Section 3, a matched molecule pair may have activity cliff on one property (dashed edges in Figure 4) but not on the other (solid edges). The graph $\mathcal{G}$ allows us to formulate molecular property prediction as a node classification problem, where the node labels describe the properties of different molecules. This graph formulation is different from those proposed by Zhuang et al. (2023) and Zhao et al. (2024), as they do not consider the AC information inside the graph (reflected by the different types of edges in Figure 4).

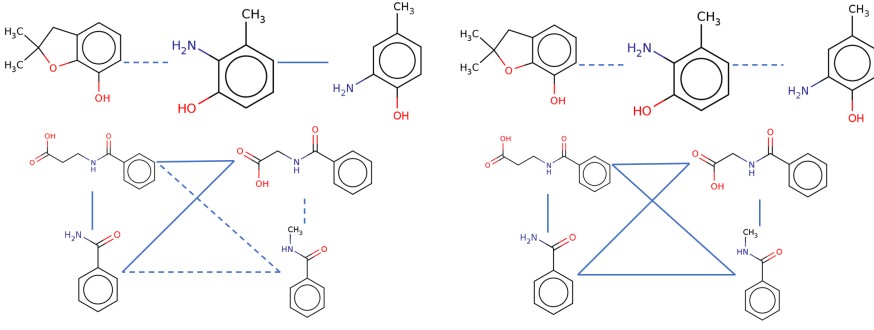

(a) Responses to ARE receptors.  (b) Responses to MMP receptors.

Figure 4: An example graph. Molecules with similar structures (as defined by Definition 3.1) are connected by edges. The edge is dashed (resp. solid) when the two molecules involved have different (resp. the same) properties.

## 4.2 NODE-LEVEL TASK FOR MOLECULES WITH ACTIVITY CLIFF

Since molecules with AC are more difficult to learn (Section 3), we consider the use of curriculum learning (Wang et al., 2022), which first selects easier samples and then harder samples to gradually train a better model. However, from Figure 3, even for molecules with similar losses, molecules with AC are still more difficult to learn than moleces without AC. As such, we propose a weighted curriculum learning algorithm that jointly considers AC and the molecule's training loss. Specifically, for a given molecule $i$, we define its weighted loss as $\hat{\ell}_i(\boldsymbol{w}) = p_i \ell_i(\boldsymbol{w})$, where $\ell_i(\boldsymbol{w})$ is the original loss on molecule $i$ (e.g., cross-entropy loss for classification tasks, or squared loss for regression

tasks), and $p_i$ is the weight on molecule $i$ defined as:

$$p_i = \begin{cases} 1 & \text{molecule } i \text{ has activity cliff} \\ p & \text{molecule } i \text{ does not have activity cliff} \end{cases} \tag{1}$$

with $p < 1$. In other words, molecules with AC have higher weights than those without AC. Thus, when two samples have the same loss values (i.e., equally difficult for the model), we select molecules with AC first. At iteration $t$, let the sampled mini-batch be $\mathcal{B}$. We select a subset $\hat{\mathcal{B}}$ of large-loss samples from $\mathcal{B}$:

$$\hat{\mathcal{B}}(\boldsymbol{w}) = \{i | i \in \mathcal{B}, \hat{\ell}_i(\boldsymbol{w}) \geq R(t) \text{ percentile of loss in } \mathcal{B}\}.$$

In other words, $R(t)$ controls the percentage of easy molecules that are discarded at iteration $t$, as we focus more on hard molecules that cannot be learned well. The loss on $\hat{\mathcal{B}}$, namely, $\mathcal{L}(\boldsymbol{w}; \hat{\mathcal{B}}(\boldsymbol{w})) = \frac{1}{|\hat{\mathcal{B}}(\boldsymbol{w})|} \sum_{i \in \hat{\mathcal{B}}(\boldsymbol{w})} \hat{\ell}_i(\boldsymbol{w})$, is then used to update the model. This allows the model to gradually focus more on the difficult molecules with AC that are more useful for making accurate prediction.

## 4.3 EDGE-LEVEL TASK FOR ACTIVITY CLIFF PAIRS

While the aforementioned sample selection method can better learn from molecules with AC, it only considers the molecules separately. However, AC is defined for a pair of molecules, and they may affect the predictions of each other. As such, we introduce an edge-level task. Specifically, for each edge $e_{ij} = (v_i, v_j)$ in $\mathcal{G}$, we define the loss:

$$\ell_{e_{ij}}(\boldsymbol{w}) = -(y_i - y_j)(\hat{y}_i(\boldsymbol{w}) - \hat{y}_j(\boldsymbol{w})), \tag{2}$$

where $y_i$ (resp., $y_j$) is the binary label for molecule $i$ (resp., $j$), and $\hat{y}_i(\boldsymbol{w})$ (resp., $\hat{y}_j(\boldsymbol{w})$) is the prediction for molecule $i$ (resp., $j$) with model parameter $\boldsymbol{w}$. For a classification task, $y_i = y_j$ indicates that molecules $i$ and $j$ have the same label and do not form an AC pair. On the other hand, when $y_i \neq y_j$, we have AC and $y_i - y_j = \pm 1$. When $y_i = 1$ (which implies $y_j = 0$), equation (2) minimizes $-(\hat{y}_i - \hat{y}_j)$, which corresponds to maximizing $\hat{y}_i$ and minimizing $\hat{y}_j$. When $y_i = 0$ (which implies $y_j = 1$), equation (2) minimizes $(\hat{y}_i - \hat{y}_j)$, which corresponds to minimizing $\hat{y}_i$ and maximizing $\hat{y}_j$. Similar deduction can be obtained for regression tasks as well, and we also draw the predictions of both molecules with activity cliff towards the ground truth. The total edge-level loss over all matched molecule pairs is then:

$$\mathcal{L}_e(\boldsymbol{w}; \mathcal{A}) = \frac{1}{|\mathcal{A}|} \sum_{e_{ij} \in \mathcal{A}} \ell_{e_{ij}} = \frac{1}{|\mathcal{A}|} \sum_{e_{ij} \in \mathcal{A}} -(y_i - y_j)(\hat{y}_i(\boldsymbol{w}) - \hat{y}_j(\boldsymbol{w})), \tag{3}$$

where $\mathcal{A} \subset \mathcal{E}$ is the set of all matched molecule pairs. The following Proposition shows the gradient of the edge-level loss in terms of the $\frac{\partial \hat{y}_i(\boldsymbol{w})}{\partial \boldsymbol{w}}$ for each molecule $i$.

**Proposition 4.1.** $\frac{\partial \mathcal{L}_e(\boldsymbol{w})}{\partial \boldsymbol{w}} = \frac{1}{|\mathcal{A}|} \sum_i -n_i(2y_i - 1) \frac{\partial \hat{y}_i(\boldsymbol{w})}{\partial \boldsymbol{w}}$, where $n_i$ is the number of AC pairs involving molecule $i$.

Proof is in Appendix A. In other words, the gradient of $\mathcal{L}_e$ is a weighted sum of $\frac{\partial \hat{y}_i(\boldsymbol{w})}{\partial \boldsymbol{w}}$'s. The weight on each $\frac{\partial \hat{y}_i(\boldsymbol{w})}{\partial \boldsymbol{w}}$ depends on the number of AC pairs involving molecule $i$, which does not change throughout training. However, not all AC pairs are equally important for the learning of discriminative molecular representations. Some pairs can be easily separated, while other pairs may be more difficult to distinguish. Thus, we also employ curriculum learning into this edge-level task, and change the edge loss in (3) to:

$$\mathcal{L}_e(\boldsymbol{w}; \hat{\mathcal{A}}) = \frac{1}{|\hat{\mathcal{A}}|} \sum_{e_{ij} \in \hat{\mathcal{A}}} \ell_{e_{ij}}(\boldsymbol{w}) = \frac{1}{R(t)|\mathcal{A}|} \sum_{e_{ij} \in \hat{\mathcal{A}}} \ell_{e_{ij}}(\boldsymbol{w}), \tag{4}$$

where $\hat{\mathcal{A}} = \{e_{ij} | e_{ij} \in \mathcal{A}, \ell_{e_{ij}} \geq R(t) \text{ percentile of loss in } \mathcal{A}\}$. Using $\hat{\mathcal{A}}$ instead of $\mathcal{A}$ allows us to focus more on AC pairs $e_{ij} \in \mathcal{A}$ with larger loss $\ell_{e_{ij}}$, which correspond to less well-separated pairs that are more important for model update.

---

**Algorithm 1** Learning with Activity Cliff (LAC).

1: Initialize prediction model $f$ with parameter $\boldsymbol{w}$ (random initialization or pre-trained weights);
2: **for** $t = 0, \ldots, T - 1$ **do**
3:     Draw a mini-batch $\mathcal{B}$ from molecule data set $\mathcal{D}$;
4:     Obtain the set $\mathcal{A}$ of molecule pairs in $\mathcal{B}$ with activity cliff;
5:     Determine $R(t)$;
6:     Select $R(t) \times |\mathcal{B}|$ large-loss samples $\hat{\mathcal{B}}$ from $\mathcal{B}$ based on network $f$'s predictions;
7:     Select $R(t) \times |\mathcal{A}|$ pairs of molecule $\hat{\mathcal{A}}$ with activity pairs and compute $\mathcal{L}_e$ in (4);
8:     Update $\boldsymbol{w} = \boldsymbol{w} - \eta \nabla_{\boldsymbol{w}}(\mathcal{L}(\boldsymbol{w}; \hat{\mathcal{B}}) + \alpha \mathcal{L}_e(\boldsymbol{w}; \hat{\mathcal{A}}))$;
9: **end for**

---

### 4.4 COMPLETE ALGORITHM

The complete algorithm, which will be called Learning with Activity Cliff (LAC), is shown in Algorithm 1. Compared with standard curriculum learning algorithms (Wang et al., 2022) that may be applied to training molecular property prediction models, it has the following two key differences: (i) Algorithm 1 involves training on two different tasks, combined together with a hyper-parameter $\alpha$, while existing works only consider curriculum learning on one task (namely the node-level task); (ii) We propose a novel design of the curriculum in Algorithm 1 based on AC information, which is unique for molecular property prediction. Note that the proposed method can be used with various (randomly-initialized or pre-trained) base models. It also introduces a hyper-parameter $R(t)$ to control the number of large-loss samples. Its effect on model performance will be studied in detail in Section 5.4.

Table 2: ROC-AUCs on various molecular property prediction classification data sets. The best performance for each task is marked in bold.

| Method | Tox21 | ToxCast | Sider | MUV | Bace | BBBP | ClinTox | HIV |
|---|---|---|---|---|---|---|---|---|
| GIN | 74.9 | 61.6 | 58.0 | 71.0 | 72.6 | 65.4 | 58.8 | 75.3 |
| GIN+LAC | **75.6** | **62.2** | **58.3** | **72.4** | **74.8** | **65.9** | **61.6** | **76.1** |
| GraphGPS | 71.5 | 68.5 | 56.4 | 66.9 | 76.9 | 67.0 | 71.1 | 77.0 |
| GraphGPS+LAC | **74.0** | **73.7** | **60.4** | **71.3** | **82.5** | **67.7** | **72.4** | **77.6** |
| GraphMVP | 75.9 | 63.1 | 63.9 | 77.7 | 81.2 | 72.4 | 79.1 | 77.0 |
| GraphMVP+LAC | **76.7** | **70.1** | **64.5** | **78.1** | **81.6** | **72.9** | **80.2** | **77.8** |
| 3D-PGT | 73.8 | 69.2 | 60.6 | 69.4 | 80.9 | 72.1 | 79.4 | 69.4 |
| 3D-PGT+LAC | **75.2** | **74.0** | **61.0** | **75.1** | **84.5** | **72.4** | **79.6** | **69.5** |
| UniMol | 79.6 | 69.6 | 65.9 | 82.1 | 85.7 | 72.9 | 91.9 | 80.8 |
| UniMol+LAC | **80.2** | **72.5** | **66.2** | **82.7** | **86.4** | **73.6** | **92.2** | **80.9** |

## 5 EXPERIMENTS

In this section, we demonstrate the performance of the proposed method on both classification data sets (Section 5.1) popularly used in existing works (Stärk et al., 2022; Wang et al., 2023b; Zhou et al., 2023) and regression data sets (Section 5.2) that are more common in real-world application (van Tilborg et al., 2022). Section 5.3 presents ablation studies to verify the effectiveness of each component in the proposed method. The effect of the hyper-parameters that define $R(t)$ are studied in Section 5.4. Section 5.5 further visualizes the loss distribution on molecules. Section 5.6 presents case studies to better understand the proposed method.

### 5.1 EXPERIMENTS ON CLASSIFICATION DATA SETS

In this section, we perform experiments on eight classification tasks from the MoleculeNet (Wu et al., 2018): Tox21, ToxCast, Sider, MUV, Bace, BBBP, ClinTox and HIV. The proposed LAC is combined with the following baseline methods: (i) training from scratch with GIN (Xu et al., 2019) and GraphGPS (Rampášek et al., 2022) models, and (ii) using model initializations from the

following pre-training methods: GraphMVP (Liu et al., 2022), 3D-PGT (Wang et al., 2023b) and UniMol (Zhou et al., 2023). Statistics of the data sets used and detailed experimental setups can be found in Appendix B.

The ROC-AUCs on various molecular property data sets are shown in Table 2. LAC improves the performance of all the models considered. With LAC, the pre-trained UniMol model achieves the best performance on all data sets except ToxCast, where 3D-PGT pre-trained model performs the best. Moreover, performance improvement depends partially on the proportion of AC samples in the data set. For example, the improvement on Tox21 is generally larger than that on MUV, and activity cliff is more commonly encountered in Tox21, as is shown in Table 8.

## 5.2 EXPERIMENTS ON REGRESSION DATA SETS

While the definition for activity cliff is straightforward for classification tasks, recent works (van Tilborg et al., 2022; Deng et al., 2023) also consider activity cliff on regression data sets. Following (van Tilborg et al., 2022), we se-

Table 3: MAE on various molecular property prediction regression data sets. The best performance for each task is marked in bold.

| Target | 5-HT1A | MOR | D3R | FXR | HRH3 |
|---|---|---|---|---|---|
| ChEMBL ID | 214 | 233 | 234 | 2047 | 264 |
| MLP(ECFP) | 0.692 | 0.845 | 0.669 | 0.796 | 0.672 |
| MLP(ECFP)+LAC | **0.656** | **0.827** | **0.635** | **0.762** | **0.657** |

lect five data sets from the ChEMBL database (Zdrazil et al., 2023), which describe the (continuous) bioactivity values of molecules to a specific target. We train a MLP with ECFP molecular fingerprints (Rogers & Hahn, 2010) as it performs best on these data sets in (van Tilborg et al., 2022), with more experimental details in Appendix B. Table 3 shows the obtained mean absolute error (MAE). The proposed LAC can also improve model performance on regression tasks.

Table 4: Ablation studies on different components in the proposed method LAC. The evaluation metric is ROC-AUC (Larger is better).

| base model | node-level curriculum | edge-level | Tox21 | ToxCast | Sider | MUV | Bace | BBBP | ClinTox | HIV |
|---|---|---|---|---|---|---|---|---|---|---|
| GraphGPS | × | × | 71.5 | 68.5 | 56.4 | 66.9 | 76.9 | 67.0 | 71.1 | 77.0 |
| | × | √ | 72.0 | 69.8 | 58.6 | 67.2 | 79.3 | 66.6 | 71.6 | 77.1 |
| | √ | × | 73.8 | 73.0 | 59.3 | 69.5 | 81.3 | 67.1 | 72.1 | 77.4 |
| | √ | √ | **74.0** | **73.7** | **60.4** | **71.3** | **82.5** | **67.7** | **72.4** | **77.6** |
| 3D PGT | × | × | 73.8 | 69.2 | 60.6 | 69.4 | 80.9 | 72.1 | 79.4 | 69.4 |
| | × | √ | 74.0 | 70.2 | 60.2 | 69.1 | 81.8 | 69.4 | 77.4 | 68.6 |
| | √ | × | 74.6 | 73.0 | **61.0** | 72.2 | 83.1 | 72.2 | **79.6** | **69.5** |
| | √ | √ | **75.2** | **74.0** | **61.0** | **75.1** | **84.5** | **72.4** | **79.6** | **69.5** |

## 5.3 ABLATION STUDIES

In this experiment, we study the effectiveness of curriculum learning in the node-level task (Section 4.2) and the pairwise loss in the edge-level task (Section 4.3). Experiments are performed on the GraphGPS model (randomly-initialized) and 3D-PGT model (pre-trained) as two representatives.

Table 5 shows the model performance with different values of $p$ in (1). Setting $p = 1$ corresponds to only using the original loss and does not distinguish molecules with/without AC, while setting $p = 0$ corresponds to using only molecules with AC for training. As can be seen, using $p < 1$ usually outperforms the baseline with $p = 1$, demonstrating the effectiveness of AC information in selecting informative molecules for training. However, setting $p$ too small can harm model performance as we neglect the congtributions of molecules without AC. In this paper, we set $p = 0.5$ as it achievs the best overall performance. Table 4 shows the ROC-AUCs obtained with or without the pairwise (edge-level) task and curriculum learning on samples (node-level curriculum). Both the pairwise task and curriculum learning on samples can generally improve the model performance, and curriculum learning on samples often has more significant improvements. The only exception is the MUV data set on 3D-PGT model, where only using pairwise task achieves slightly worse performances. That

is because MUV data set contains fewer molecules with activity cliff, as is shown in Table 8 in Appendix B. Combining both components achieves the best overall performances across all data sets.

Table 5: Ablation studies on the effect of activity cliff weights for curriculum learning on samples. The evaluation metric is ROC-AUC (Larger is better).

| base model | $p$ | Tox21 | ToxCast | Sider | MUV | Bace | BBBP | ClinTox | HIV |
|---|---|---|---|---|---|---|---|---|---|
| | 1.0 | 73.5 | 72.6 | 60.3 | **72.9** | 80.0 | 65.0 | 71.1 | 77.0 |
| | 0.75 | 73.8 | 72.9 | 60.4 | 72.3 | 81.7 | 67.3 | 71.9 | 77.5 |
| GraphGPS | 0.5 | **74.0** | **73.7** | **60.4** | 71.3 | **82.5** | **67.7** | **72.4** | **77.6** |
| | 0.25 | 71.6 | 70.3 | 57.9 | 69.8 | 77.4 | 67.1 | 70.1 | 73.4 |
| | 0 | 67.8 | 66.9 | 56.3 | 69.2 | 75.8 | 66.3 | 67.8 | 72.2 |
| | 1.0 | 74.2 | 73.0 | 60.7 | 72.9 | 81.5 | 70.5 | 79.4 | 69.4 |
| | 0.75 | 74.7 | 73.7 | 60.9 | 74.6 | 83.8 | 72.1 | **79.6** | 69.2 |
| 3D PGT | 0.5 | **75.2** | **74.0** | **61.0** | **75.1** | **84.5** | **72.4** | **79.6** | **69.5** |
| | 0.25 | 72.4 | 71.9 | 59.2 | 70.4 | 81.3 | 71.8 | 73.6 | 69.1 |
| | 0 | 68.6 | 68.9 | 58.6 | 64.6 | 79.1 | 65.7 | 69.1 | 68.7 |

Table 6: ROC-AUC on different data sets with different types of $R(t)$ schedules. 3D-PGT pre-trained model is used.

| Schedule | Tox21 | ToxCast | Sider | MUV | Bace | BBBP | ClinTox | HIV |
|---|---|---|---|---|---|---|---|---|
| linear | **75.2** | **74.0** | **61.0** | **75.1** | 84.5 | **72.4** | **79.6** | **69.5** |
| root | 74.5 | 73.2 | 59.5 | 71.0 | 83.5 | 70.0 | 79.3 | 69.2 |
| geometric | 75.0 | 73.7 | 60.5 | 75.0 | **85.0** | 71.2 | **79.6** | 69.3 |

## 5.4 IMPACTS OF $R(t)$

In this section, we investigate how different $R(t)$ schedules (in Algorithm 1) affects the performance of LAC. We consider the following three schedules: (i) linear: $R(t) = \lambda \min(t/(\gamma T), 1)$, which increases the difficulty of training samples at a uniform rate; (ii) root: $R(t) = \lambda \min(\sqrt{t/(\gamma T)}, 1)$, which introduces more *hard* samples in fewer epochs; and (iii) geometric: $R(t) = \lambda(2^{\min(t/(\gamma T), 1)} - 1)$, which

Table 7: ROC-AUC on Tox21 data set with different $\lambda$ and $\gamma$ for LAC. 3D-PGT pre-trained model is used.

| | $\lambda=0.1$ | $\lambda=0.2$ | $\lambda=0.3$ | $\lambda=0.4$ | $\lambda=0.5$ |
|---|---|---|---|---|---|
| $\gamma=0.1$ | **75.1** | **75.2** | **75.1** | 73.8 | **75.2** |
| $\gamma=0.2$ | 74.0 | **75.0** | **75.1** | 72.7 | 74.3 |
| $\gamma=0.3$ | 73.0 | **75.0** | **75.0** | 72.3 | 72.2 |
| $\gamma=0.4$ | 73.7 | 73.1 | 74.1 | 72.1 | 72.2 |
| $\gamma=0.5$ | 74.6 | 72.5 | 72.3 | 72.6 | 73.5 |

trains for a greater number of epochs on the subset of *easy* samples. We set $\gamma = 0.1$ and $\lambda = 0.2$ for all schedules. Table 6 shows the ROC-AUCs obtained on the various data sets. In general, the linear schedule achieves the best performance, and the root schedule achieves the worst performances.

Using the linear schedule, Table 7 shows the ROC-AUCs on the Tox21 data set with different hyper-parameters $\gamma$ and $\lambda$. The performance is stable on a wide range of $\gamma$ and $\lambda$ values.

## 5.5 LOSS DISTRIBUTIONS FOR MOLECULES WITH ACTIVITY CLIFF

In this section, we compare the training loss distributions on molecules with AC obtained with and without the proposed LAC. Figure 5 (resp. Figure 6) shows the distributions with the randomly-initialized GraphGPS model (resp. pre-trained 3D-PGT model) at the end of the training process. Models trained by the baseline algorithm (blue columns) have inaccurate predictions on part of molecules with AC, while the proposed method LAC (orange columns) can reduce the loss for these samples. LAC improves the performance for both randomly-initialized model and pre-trained model.

## 5.6 CASE STUDIES

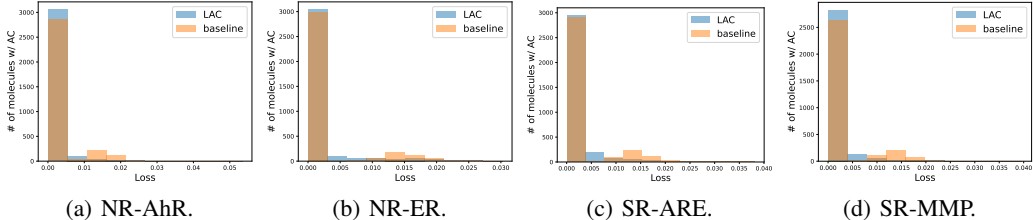

(a) NR-AhR.    (b) NR-ER.    (c) SR-ARE.    (d) SR-MMP.

Figure 5: Loss distributions obtained by the randomly-initialized GraphGPS model on molecules with AC on 4 tasks in Tox21.

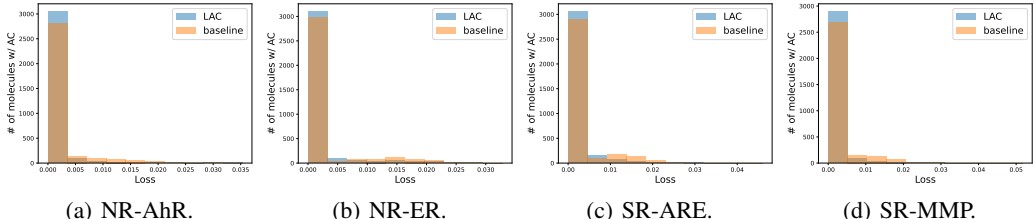

(a) NR-AhR.    (b) NR-ER.    (c) SR-ARE.    (d) SR-MMP.

Figure 6: Loss distributions obtained by 3D-PGT pre-trained GraphGPS model on molecules with AC on 4 tasks in Tox21.

Finally, we choose some examples to illustrate how the proposed method LAC can improve upon existing molecular property prediction model. We choose the UniMol pre-trained model as it achieves the best overall performance on various data sets. As in Figure 7(b), without the proposed LAC, UniMol cannot correctly classify molecules with AC when the structural differences are very small, even if it can handle easier pairs like in Figure 7(a). With tasks from two levels,

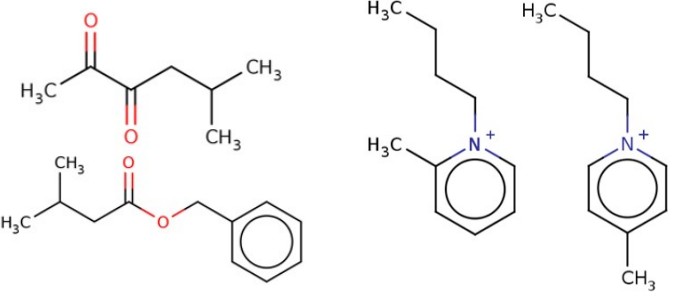

(a) Example 1: correctly classified by both UniMol and UniMol+LAC

(b) Example 2: wrongly classified by UniMol but correctly classified by UniMol+LAC

Figure 7: Examples of molecules with AC. LAC improves upon existing methods to obtain more accurate predictions on molecules with AC.

LAC further improves the model performance to accurately classify two molecules in Figure 7(b)

## 6 CONCLUSION

In this paper, we propose to improve the performance of molecular property prediction models from the perspective of activity cliff (AC). We first use empirical results with different tasks and models to demonstrate that standard training pipeline cannot learn molecules with AC well. By reformulating the original problem as a graph problem, we propose a novel training algorithm LAC that uses both node-level and edge-level tasks to effectively learn from molecules with AC. Extensive empirical results demonstrate that the proposed method significantly improves the performance of different baseline methods.

### ACKNOWLEDGEMENTS

Q. Yao's work is supported by National Natural Science Foundation of China (under Grant No. 92270106) and Beijing Natural Science Foundation (under Grant No. 4242039). This is also supported in part by the Research Grants Council of the Hong Kong Special Administrative Region (Grants 16200021, 16202523 and C7004-22G-1).

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

## A  PROOF OF PROPOSITION 4.1

*Proof.*

$$\frac{\partial \mathcal{L}_e}{\partial \boldsymbol{w}} = \frac{1}{|\mathcal{A}|} \sum\nolimits_{e_{ij} \in \mathcal{A}} \frac{\partial \ell_{e_{ij}}}{\partial \boldsymbol{w}}$$
$$= \frac{1}{|\mathcal{A}|} \sum\nolimits_{e_{ij} \in \mathcal{A}} -(y_i - y_j)(\frac{\partial \hat{y}_i}{\partial \boldsymbol{w}} - \frac{\partial \hat{y}_j}{\partial \boldsymbol{w}})$$

Regarding $y_i - y_j$, since $e_{ij} \in \mathcal{A}$ indicates that molecule $i$ and $j$ have activity cliff, we consider the following two cases:

- $y_i = 1$, then we must have $y_j = 0$, and $y_i - y_j = 1 = 2y_i - 1$

- $y_i = 0$, then we must have $y_j = 1$, and $y_i - y_j = -1 = 2y_i - 1$

Thus, we have $y_i - y_j = 2y_i - 1$ always holds and:

$$\frac{\partial \mathcal{L}_e(\boldsymbol{w})}{\partial \boldsymbol{w}} = \frac{1}{|\mathcal{A}|} \sum\nolimits_{i:e_{ij} \in \mathcal{A}} -(2y_i - 1)\frac{\partial \hat{y}_i}{\partial \boldsymbol{w}}$$
$$= \frac{1}{|\mathcal{A}|} \sum\nolimits_i -n_i(2y_i - 1)\frac{\partial \hat{y}_i}{\partial \boldsymbol{w}}.$$

$\square$

## B  EXPERIMENTAL DETAILS

All experiments are run on a single NVIDIA RTX A6000 GPU. For all experiments in this work, we use the Adam optimizer (Kingma & Ba, 2015), and follow its default hyper-parameters: learning rate $\eta$ is set 0.001, first-order momentum weight $\beta_1$ is set to 0.9, and the second-order momentum weight $\beta_2$ is set to 0.99. The batch size is set to 256 for all data sets.

Unless otherwise specified, we set the $R(t)$ schedule as $R(t) = \lambda \min(t/(\gamma T), 1)$ with $\lambda = 0.2$ and $\gamma = 0.1$, and the weight $\alpha$ for pairwise loss $\mathcal{L}_e$ is set to 0.1. For the classification experiments, the data splits of all data sets in our experiments follow the scaffold split in (Wang et al., 2023b). For the regression experiments, the data splits of all data sets in our experiments are the same as in (van Tilborg et al., 2022), and we use an three-layer MLP model with input dimension 1024 and hidden dimension 512 for all hidden layers.

All data sets used in our experiments are released under MIT license. Some statistics on data sets used in experiments are in Table 8 (classification) and Table 9 (regression).

Table 8: Summary for the data sets used for classification tasks.

|  | Tox21 | ToxCast | Sider | MUV | Bace | BBBP | ClinTox | HIV |
|---|---|---|---|---|---|---|---|---|
| # molecules | 7831 | 8521 | 1427 | 93087 | 1513 | 2039 | 1477 | 41127 |
| # MMPs | 3212114 | 3802710 | 11935 | 2243595 | 15894 | 24105 | 7080 | 20740266 |
| # AC pairs | 315841 | 381260 | 3183 | 2610 | 1470 | 1186 | 1912 | 2484912 |
| AC ratio (%) | 9.83 | 10.03 | 26.67 | 0.12 | 9.23 | 4.92 | 27.01 | 11.98 |

## C  ADDITIONAL EMPIRICAL RESULTS

### C.1  ABLATION STUDY ON THE IMPACT OF CURRICULUM LEARNING FOR PAIRWISE TASK

Table 10 compares the model performances on whether to use curriculum learning for pairwise task. We see that using curriculum learning for pairwise task further improves the performance than using the naive pairwise task for most data sets.

Table 9: Summary for the data sets used for regression tasks.

|            | 5-HT1A | MOR   | D3R   | FXR   | HRH3  |
|------------|--------|-------|-------|-------|-------|
| # molecules| 3317   | 3142  | 3657  | 3097  | 2862  |
| # MMPs     | 19240  | 17200 | 26707 | 21264 | 15652 |
| # AC pairs | 6734   | 6045  | 10418 | 9282  | 5913  |
| AC ratio (%)| 35.00 | 35.15 | 39.01 | 43.65 | 37.78 |

Table 10: Ablation studies on the effect of curriculum learning for pairwise task. The evaluation metric is ROC-AUC (Larger is better).

| Base model | Pairwise curriculum | Tox21 | ToxCast | Sider | MUV | Bace | BBBP | ClinTox | HIV |
|------------|---------------------|-------|---------|-------|------|------|------|---------|------|
| GraphGPS   | ×                   | 73.0  | 73.2    | 59.7  | 70.6 | 81.6 | **67.8** | 71.9 | 77.5 |
|            | √                   | **74.0** | **73.7** | **60.4** | **71.3** | **82.5** | 67.7 | **72.4** | **77.6** |
| 3D PGT     | ×                   | 74.0  | 73.5    | 60.8  | 71.0 | 83.6 | 70.4 | 78.9 | 69.1 |
|            | √                   | **75.2** | **74.0** | **61.0** | **75.1** | **84.5** | **72.4** | **79.6** | **69.5** |

## C.2 EFFECTS OF BATCH SIZE

Table 11 compares the performance with different batch sizes for LAC on both GraphGPS and 3D-PGT model. While we set the batch size to be 256 for all data sets in our experiments, we can see that setting the batch size either too large (1024) or too small (64) may not lead to the best performance. Setting the batch size too small cannot cover enough activity cliff pairs in the edge-level loss of our method, hence cannot utilize this task well and may even leads to performance worse than the standard training pipeline (e.g., the Tox21 data). While setting the batch size larger leads to some improvement on large data sets like MUV or ToxCast, it leads to even worse performance for other data sets with limited molecules like Sider or BBBP. Such observation agrees with existing theoretical works on stochastic optimization for neural networks (Lin et al., 2018; 2020), as they demonstrate that large batch sizes can lead to worse generalization performance. Therefore, although setting the batch size to be larger can include more activity cliff pairs in a single batch, it may still not lead to better performance on all data sets.

Table 11: Ablation studies on the effect of batch size. The evaluation metric is ROC-AUC (Larger is better).

| Method | Batch size | Tox21 | ToxCast | Sider | MUV | Bace | BBBP | ClinTox | HIV |
|--------|-----------|-------|---------|-------|------|------|------|---------|------|
| GraphGPS+LAC | 64   | 72.9  | 72.1    | 59.7  | 70.7 | 81.9 | 67.1 | 72.2 | 77.3 |
|              | 256  | **74.0** | 73.7 | **60.4** | 71.3 | **82.5** | **67.7** | **72.4** | **77.6** |
|              | 1024 | 73.9  | **73.8** | 58.2  | **71.6** | 81.5 | 66.4 | 71.7 | 77.4 |
| 3D PGT+LAC   | 64   | 74.9  | 73.8    | 60.5  | 73.9 | 83.8 | 72.1 | 79.3 | 69.1 |
|              | 256  | **75.2** | **74.0** | **61.0** | 75.1 | **84.5** | **72.4** | **79.6** | **69.5** |
|              | 1024 | 75.0  | **74.0** | 60.1  | **75.2** | 81.3 | 71.8 | 76.6 | 69.2 |

## C.3 TIME COST ON ACTIVITY CLIFF DETECTION

Table 12 compares the total time cost in fine-tuning for the standard training pipeline and our proposed method LAC. Note that compared to standard training, LAC involves an additional process of finding all activity cliff pairs, therefore we show its time cost in two parts in parenthesis, where the first number represents the time cost of finding all activity cliff pairs and the second number represents the time cost of fine-tuning in Algorithm 1. We can see that the time cost for our method is almost the same as the standard training pipeline. In other words, the new node and edge-level tasks do not incur much additional time cost. Also, the time cost of finding all activity cliff pairs is generally limited compared to fine-tuning.

Table 12: CPU time cost (in minutes) of standard training pipeline and the proposed method LAC when fine-tuning 3D-PGT/UniMol model.

| Data sets | Tox21 | Sider | Bace | BBBP |
|---|---|---|---|---|
| 3D-PGT | 156 | 54 | 62 | 69 |
| 3D-PGT+LAC | 196 (37+159) | 58 (3+55) | 70 (5+65) | 73 (4+69) |
| UniMol | 208 | 77 | 83 | 91 |
| UniMol+LAC | 247 (37+210) | 80 (3+77) | 88 (5+83) | 97 (4+93) |

## C.4 ENLARGED FIGURES IN SECTION 3 AND ADDITIONAL MOTIVATION RESULTS

Figure 8 shows the average training losses for molecules with AC and molecules without AC. As can be seen, for all the four setups, **molecules with AC have significantly larger training losses than molecules without AC**. This demonstrates that molecules with AC are more difficult to learn due to their similar structures yet different properties. Moreover, from Figures 8(c) and 8(d), we can see that this phenomenon also exists for the 3D-PGT and Uni-Mol pre-trained models. In other words, molecules with AC are still more difficult to learn during fine-tuning of pre-trained models.

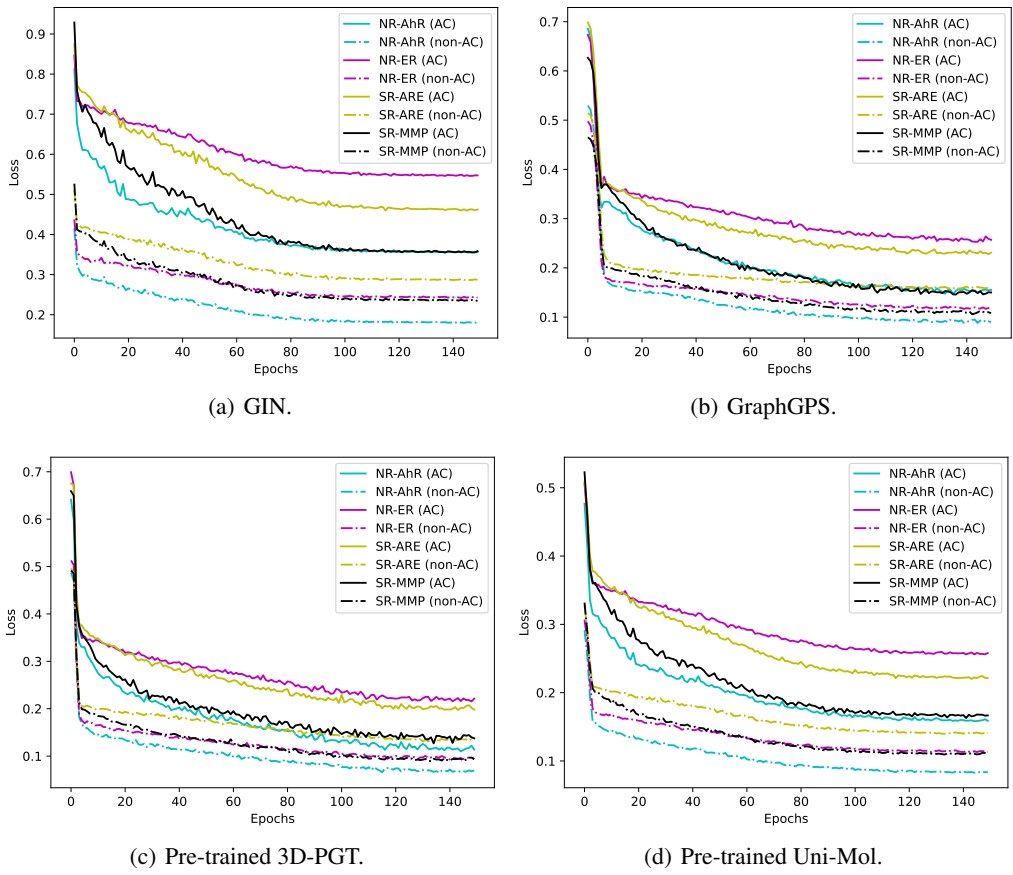

(a) GIN.

(b) GraphGPS.

(c) Pre-trained 3D-PGT.

(d) Pre-trained Uni-Mol.

Figure 8: Training losses of molecules with and without activity cliffs in four model training setups.

Since a pair of molecules with activity cliff have large difference in their properties, they may have larger influence on the prediction of each other during training. To demonstrate this, Figure 9 shows the average difference of training losses ("loss gap") between molecules with activity cliff. As can be seen, **AC leads to loss gaps between two molecules**, which also indicates that all these models fail to accurately classify both molecules, as in such cases the loss gap should be small (both with small loss). Instead, current models make the same prediction for these two molecules with AC. Only one

molecule is correctly classified with small loss, while another molecule has large loss that leads to the large loss gap.

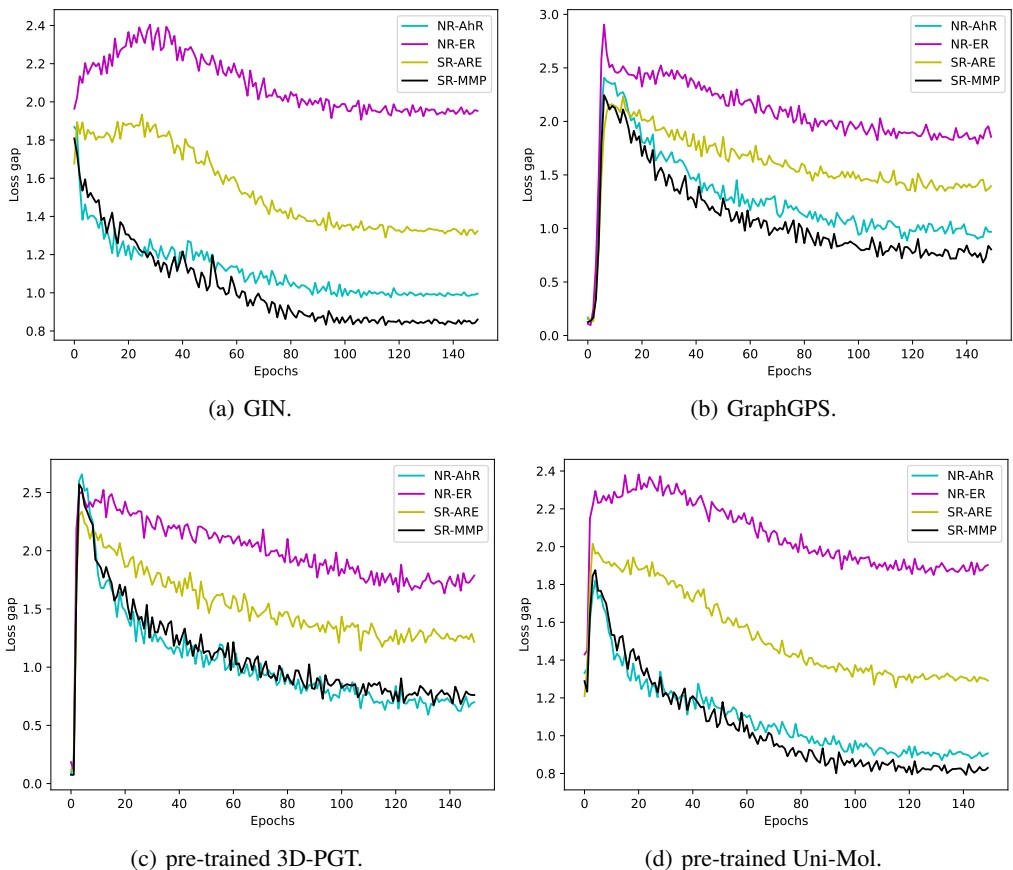

(a) GIN.

(b) GraphGPS.

(c) pre-trained 3D-PGT.

(d) pre-trained Uni-Mol.

Figure 9: Loss gaps of molecules with AC for different tasks during model training.

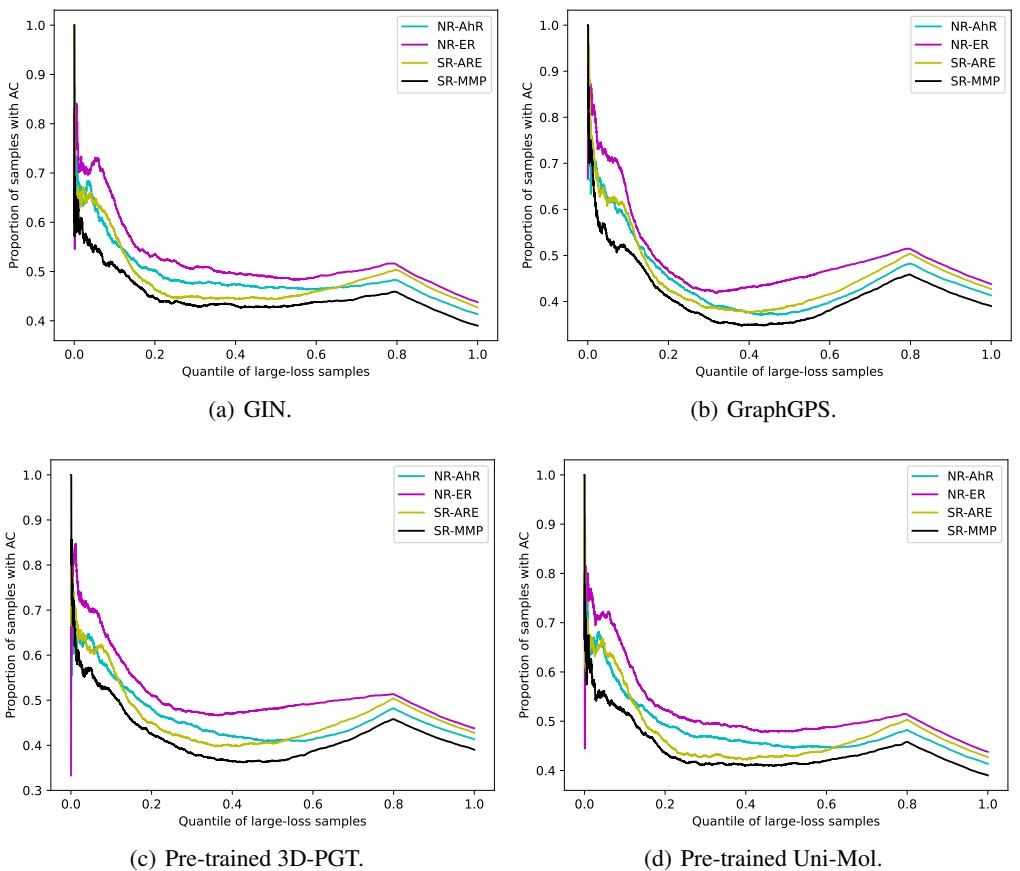

(a) GIN.

(b) GraphGPS.

(c) Pre-trained 3D-PGT.

(d) Pre-trained Uni-Mol.

Figure 10: (Larger version of Figure 2) Proportion of molecules with AC among molecules with top-$n\%$ loss values.

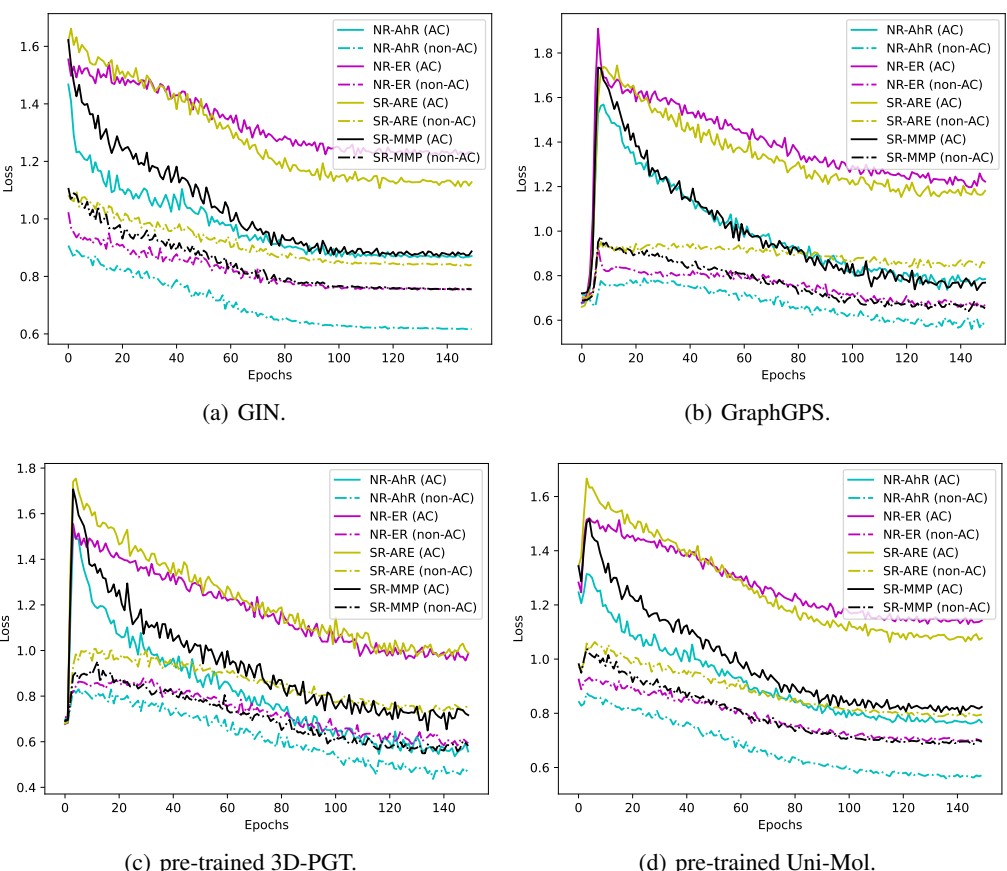

(a) GIN.

(b) GraphGPS.

(c) pre-trained 3D-PGT.

(d) pre-trained Uni-Mol.

Figure 11: (Larger version of Figure 3) Training losses of large-loss molecules with and without activity cliffs in four model training setups.

