# OpenReview forum: "Curriculum-aware Training for Discriminating Molecular Property Prediction Models"
_ICLR.cc/2025/Conference — ICLR 2025 Poster_

### Official Review · Reviewer_nmrm · 2024-10-25

**Soundness:** 3
**Presentation:** 3
**Contribution:** 3
**Rating:** 8
**Confidence:** 5

**Summary:**

This paper proposes a curriculum learning-based training method (LAC) for molecular graph learning on property prediction tasks. Empirical evidences are provided to expose the limitation of current molecular graph learning is the insufficient learning for activity cliff molecules. Further, an elaborate definition is designed to convert the general graph learning to node-level and edge-level tasks with activity cliff being considered, followed by a designed integrated loss function. The experiments on several general molecular property prediction tasks show improvements of LAC than ordinary training method.

**Strengths:**

- The empirical analysis results effectively highlight the current limitations in molecular property prediction, especially concerning the activity cliff issue.

- The definitions of node-level and edge-level tasks, considering activity cliffs, are clearly elaborated.

**Weaknesses:**

- It’s unclear why only an MLP model was applied for regression tasks but other models (GraphGPS, GraphMVP, etc.) were excluded. Additionally, the choice of specific ChEMBL assay data (please include the ChEMBL ID) over commonly used molecular property regression benchmarks (e.g., FreeSolv, ESOL) is not explained.

- A study on curriculum learning for molecular graph learning and property prediction should be referenced to strengthen the related works section:

Ref:
Gu Y, Zheng S, Xu Z, et al. An efficient curriculum learning-based strategy for molecular graph learning[J]. Briefings in Bioinformatics, 2022, 23(3): bbac099.

- A comparison of computation times with standard random-sampling training would provide additional context for performance evaluation regarding the time cost issue.

- LAC demonstrates performance improvements, it would be helpful to clarify how these gains are achieved—does LAC contribute more to activity cliff (AC) data, non-AC data, or both? A table or figure result may be helpful to understand that.

- Since the loss functions for standard training and curriculum learning differ in equations and terms, directly comparing their values may be misleading. An alternative approach would be to show the proportions of large-loss instances for AC data across each training strategy throughout the process (e.g., similar to Figure 3).

**Questions:**

- What are the criteria used for the detection of activity cliffs? Since it is a concept mainly for binding affinity, people may be more interested about how the authors transfer and expand such concept to molecular property tasks (especially regression tasks and some non-affinity property tasks such as BBBP) with rationales behind. Please provide more details about the clear criteria and other necessary descriptions about this. It could be the most important experimental setting and basis of the study to ensure accurate definition for AC is applied.

- How is the alpha ratio for pairwise loss determined?

- Some typos exist. Such as "ChemBL" should be "ChEMBL".

---

> ### Author Response · Authors · 2024-12-01
> **Responses to reviewer nmrm**
>
> We would like to first thank you for your recoginition of our work as well as your valuable suggestions. Here we reply to your comments point by point:
>
> > **W1-1.** It's unclear why only an MLP model was applied for regression tasks but other models (GraphGPS, GraphMVP, etc.) were excluded.
>
> We choose the MLP model as it has been used on the ChEMBL database in existing
> works on activity cliff (van Tilborg et al. 2022) and it generally achieves satisfying performance.
> This is clarified in
> Section 5.2 of
> this revised version.
>
> > **W1-2.** Additionally, the choice of specific ChEMBL assay data (please include the ChEMBL ID) over commonly used molecular property regression benchmarks (e.g., FreeSolv, ESOL) is not explained.
>
> We selected these assay data as they have been identified as influenced by AC in (van Tilborg et al. 2022).
> As suggested,
> we now include the ChEMBL ID in Table 3 of Section 5.2.
>
> > **W2.** A study on curriculum learning for molecular graph learning and property
> prediction should be referenced
> to strengthen the related works section
>
> Thank you for your suggestion.
> We have added
> your suggested
> reference
> and more discussions
> in Section 2.2
> (highlighted in blue).
>
> > **W3.** A comparison of computation times with standard random-sampling training would provide additional context for performance evaluation regarding the time cost issue.
>
> The comparison depends on whether
> the dataset with activity cliff pairs is available.
> When this dataset is available (e.g., as is provided in
> (van Tilborg et al., 2022)),
> the proposed method
> (Algorithm 1)
> has almost the same time cost as
> standard random-sampling training,
> as is demonstrated in the new
> Table 13 (in Appendix C.3 of this revised version).
> However, when this data set is not available, currently we use a brute-force approach to examine all molecule pairs, which takes time quadratic in the number of molecules
> (as can also be seen from Table 13).
> More effective algorithms to find activity cliff pairs from a given molecular
> property prediction dataset is beyond the scope of this work,
> and will be addressed
> in the future.
>
> > **W4.** LAC demonstrates performance improvements, it would be helpful to clarify how these gains are achieved—does LAC contribute more to activity cliff (AC) data, non-AC data, or both? A table or figure result may be helpful to understand that.
>
> Thank you for your suggestion.
> In this revised version,
> we add
> a new Table 11 in Appendix C.1 that compares the ROC-AUCs on
> molecules
> with and without AC
> by the standard training pipeline and proposed LAC
> (using the Uni-Mol model).
> As expected, LAC yields a bigger
> improvement on molecules with AC.
>
> > **W5.** Since the loss functions for standard training and curriculum learning differ in equations and terms, directly comparing their values may be misleading. An alternative approach would be to show the proportions of large-loss instances for AC data across each training strategy throughout the process (e.g., similar to Figure 3).
>
> We suppose the reviewer is referring to the visualization of loss distributions in
> Figures 5 and 6.
> Note that while the
> training loss for curriculum learning
> is different
> from that of standard training (as is highlighted in blue in Section 4.2),
> the plots
> in Figures 5-6
> are based
> on the cross-entropy loss for both the baseline and LAC, and thus
> the comparison is fair.
> This is now clarified
> (highlighted in blue) in
> Section 5.5 of this revised version.
>
> > **Q1.** What are the criteria used for the detection of activity cliffs?
>
> As stated in Definition 3.2,
> activity cliff refers to a matched molecule pair
> (defined in Definition 3.1) with different labels on a given property.
> For classification tasks,
> the "label" simply refers to the class labels.
> For regression tasks, we follow (van Tilborg et al., 2022)  and consider
> the two molecules have
> different labels
> when the target value of
> one molecule is
> at least 11 times larger than that of the other.
>
> > **Q2.** How is the alpha ratio for pairwise loss determined?
>
> As is mentioned in Appendix B, we set $\alpha=0.1$ for all experiments.
>
> > **Q3.** Some typos exist. Such as "ChemBL" should be "ChEMBL".
>
> Thank you for your pointing this out. We have thoroughly revised our submission
> and correct the spelling errors.

---

> > ### Comment · Reviewer_nmrm · 2024-12-01
> > **Comments to the authors' responses.**
> >
> > Thank you for addressing most of my concerns. I am increasing my score to 8. However, I would still recommend that the authors consider adding a brief analysis or summary of the limitations and potential directions for future work in the manuscript.
> >
> > Specifically, some limitations and drawbacks remain unaddressed. For instance, the time cost and feasibility of the method when data is unavailable for practical usage warrant further discussion. Additionally, the experiments for regression tasks could be expanded. While the current tasks focus on bioactivity prediction, which understandably highlights the impact of activity cliffs, other well-recognized and general benchmarks but only weakly related to activity cliffs (e.g., FreeSolv, ESOL) have not been considered.

---

> > > ### Author Response · Authors · 2024-12-03
> > > **Thanks and some further responses**
> > >
> > > We are glad to know that our responses addressed most of your previous concerns.
> > > Since the deadline for updating our submission has passed,
> > > we promise to add some discussion on the time cost issue of LAC as its possible limiation in our final version.
> > > We will also consider adding more experiments on other regression data sets (e.g., FreeSolv, ESOL) you have mentioned.
> > > Thank you again for your recognition of our work and increasing your rating score!

---

### Official Review · Reviewer_YTtN · 2024-10-30

**Soundness:** 3
**Presentation:** 3
**Contribution:** 3
**Rating:** 8
**Confidence:** 4

**Summary:**

------- Update -------

Thank you to the authors for addressing the comments and questions I had on the paper, I'd like to say again what a pleasure it was to read and that I really appreciate both the content clarifications and updates to the presentation.

On reviewing your answers to my questions and reading the paper again, I'm very happy to raise my score to an 8.
Well done on the paper, and thank you again for all of your hard work.



-----------
Thank you for a really interesting read. I found both the analysis of model performance on the molecules with activity cliffs a strong addition to the literature, to take something commonly acknowledged and quantify it is vital work, and the novel exploration of the problem very good to see.

Summary:
* The authors present a novel approach to tackling the problem of molecules exhibiting an activity cliff. This problem plagues computational chemists as many molecules share large portions of their structure, differing only for a small percentage of the molecule, and still have very different properties. Such properties are intuitive to human experts, but hard to handle using many deep learning methods.
* The authors take this anecdotal knowledge and make strong baseline measurements of the discrepancy found in many SOTA models in cases where activity cliffs are present.
* The authors then reformulate the problem of molecular property prediction into a node classification problem as part of a graph structure representing molecules with similar structures.
* They present this worm as Learning with Activity Cliff (LAC) - and use the concept of curriculum learning to slowly introduce harder to separate AC molecules - this work shows improvements across a range of benchmark datasets and conduct extensive ablation studies to investigate which properties are contributing.

**Strengths:**

* The analysis of the well known but poorly quantified impact  of molecules with activity cliffs is a really valuable contribution to the field
* The novel problem construction looks like a genuinely different way to approach training models for molecular property prediction.
* The extensive ablation study is really strong showing which components of the restructuring are contributing to the improvements in score - this means as a reader I have the ability to start replicating / incorporating this method into new work.
* The combination of new reformatting of the problem with the curriculum learning approach to slowly introduce more complex examples seems like a very promising path.

**Weaknesses:**

* The degree of improvement over the baselines in each case was perhaps hard to quantify - I really appreciate the comparison (Table 2, 3) with the baseline models and + LAC - but felt these tables perhaps lacked context. From the tables alone it’s hard to evaluate if the degree of improvement is significant or not. If a couple of reference models could be added to these tables to show other work that would help calibrate my perception of the improvement of the LAC method.
* The comment about the lack of baseline against which to judge the contributions applies to tables 4 and 5 and 6 as well. (However I recognise that adding / evaluating baseline models for all cases can be expensive / difficult to conduct.)
* The way the initial node features are generated feels unclear to me - on line 228 “In this graph, each molecule corresponds to a node, and the molecule’s chemical structure can be stored as node features” - How exactly are these features chosen? Are these features the readout from the baseline models (GraphGPS, UniMol ?) or are these simply generated from something like RDKit. My interpretation from the text is the former, but some more clarity here in the text would be good.
* I find it slightly unclear what the exact message is - my take away is the LAC is a really good way to improve the abilities of an already performed model - but I'm left concluding the paper a bit unsure exactly how strong the case is.
* The loss distributions in Fig 6 are very hard to read, please use the same binning scheme for both histograms and show them either with low alpha or as lines only so I can see how they change. Also the text is too small, try adjusting figure sizes for these results?

**Questions:**

Questions / Suggestions:
* What is the impact of the batch size on training here? In line 3 of the algorithm (line 312) the mini batch is chosen, then pairs with the activity cliff found. This means the second term in the loss L_e is dependant on how many samples are found, did you study the impact of the batch size on the training? As a reader I would want to know - my dataset has X% of possible activity cliff molecules, what batch size Y do I need to see an improvement of magnitude Z using this method? Otherwise I would suspect the impact to only be a small reguarlising term? Is this a correct analysis - and could some more detail be given on this point?
* Do you have any analysis at how effectively the LAC process picks out the molecules which have an activity cliff? It selects the ones with high loss and then forms edges based on the relative labels - but do you know how much of the time these pairs do form AC pairs?


Formatting - lower priority but would be good to aesthetically improve
* Figure text sizes - the text on many of the figures is unreadable at a normal zoom level, try adjusting the matplotlib params to make these more readable. You can always move some into the appendix and leave one or two examples from each figure in the main text. Specially Fig 2, 3 and 6
* Caption of Fig 7 is a bit unwieldy
* Line 194 has a typo “even they” -> “even though they” ?


Thank you again for the work, I found the paper really enjoyable to read and showed strong scientific process.
Some clarifying on a few points and tidying up of some of the figures are my main concerns, otherwise I find the work very solid.

---

> ### Author Response · Authors · 2024-12-01
> **Responses to reviewer YTtN**
>
> We would like to first thank you for your recoginition of our work as well as your valuable suggestions. We are glad to know that you think this paper really enjoyable to read and show strong scientific process. Here we reply to your comments point by point:
>
> > **W1.** The degree of improvement over the baselines in each case was perhaps hard to quantify - I really appreciate the comparison (Table 2, 3) with the baseline models and + LAC - but felt these tables perhaps lacked context. From the tables alone it’s hard to evaluate if the degree of improvement is significant or not. If a couple of reference models could be added to these tables to show other work that would help calibrate my perception of the improvement of the LAC method.
>
> To the best of our knowledge, no existing work has considered improving the performance of molecular property prediction models from the activity cliff perspective, and Table 2 and 3 demonstrate that the proposed method LAC consistently improves the prediction accuracy for different pre-trained models on different data sets.
>
> > **W2.** The comment about the lack of baseline against which to judge the contributions applies to tables 4 and 5 and 6 as well. (However I recognise that adding / evaluating baseline models for all cases can be expensive / difficult to conduct.)
>
> Tables 4-6 are ablation studies that show how different components in the proposed method contribute to the improved performance. Note that we used two base models (GraphGPS and 3D PGT). Table 4 compares the effects of using/not using the node-level loss and the edge-level loss. Table 5 shows the effects of weight $p$ in equation (1). Table 6 shows the effect of using curriculum learning on the edge-level loss.
>
> > **W3.** The way the initial node features are generated feels unclear to me - on line 228 “In this graph, each molecule corresponds to a node, and the molecule’s chemical structure can be stored as node features” - How exactly are these features chosen? Are these features the readout from the baseline models (GraphGPS, UniMol ?) or are these simply generated from something like RDKit. My interpretation from the text is the former, but some more clarity here in the text would be good.
>
> Thank you for mentioning this. We directly use the readout from baseline models as node features. This is now clarified in Section 4.1 of this revised version (highlighted in blue).
>
> > **W4.** I find it slightly unclear what the exact message is - my take away is the LAC is a really good way to improve the abilities of an already performed model - but I'm left concluding the paper a bit unsure exactly how strong the case is.
>
> Empirical results in Section 3 demonstrate that existing models all fail to effectively tackle the challenge of activity cliff. On the other hand, the proposed LAC can help improve the performance of various molecular property prediction models by training from the hard molecules more effectively.
>
> > **W5.** The loss distributions in Fig 6 are very hard to read, please use the same binning scheme for both histograms and show them either with low alpha or as lines only so I can see how they change. Also the text is too small, try adjusting figure sizes for these results?
>
> Thank you for your suggestion. In this revised version, we now use the same binning scheme for both methods. Moreover, we now use lower alpha value and larger text size in Figures 5 and 6 for easier comparison between theh baseline and our method LAC.
>
> > **Q1.** What is the impact of the batch size on training here? In line 3 of the algorithm (line 312) the mini batch is chosen, then pairs with the activity cliff found. This means the second term in the loss L_e is dependant on how many samples are found, did you study the impact of the batch size on the training? As a reader I would want to know - my dataset has X% of possible activity cliff molecules, what batch size Y do I need to see an improvement of magnitude Z using this method? Otherwise I would suspect the impact to only be a small reguarlising term? Is this a correct analysis - and could some more detail be given on this point?
>
> As suggested, we have conducted additional experiments on the influence of batch size. Table 12 in Appendix C.2 shows the ROC-AUC on different data sets with different batch sizes for LAC on both the GraphGPS and 3D-PGT models. From this table, we can see that an intermediate batch size of 256 often works well. A very small batch size is undesirable as the number of sample pairs is limited and makes
> the edge-level loss less useful. On the other hand, while a very large batch size can be useful for some large data sets (e.g., MUV or ToxCast), theoretical results on stochastic optimization [1,2] show that the model may converge to a sharp minimum with poor generalization performance when a large batch size is used.
>
> [1] Don't Use Large Mini-Batches, Use Local SGD. ICLR 2018
>
> [2] Extrapolation for Large-batch Training in Deep Learning. ICML 2020

---

> > ### Author Response · Authors · 2024-12-01
> > **Responses to reviewer YTtN (cont.)**
> >
> > > **Q2.** Do you have any analysis at how effectively the LAC process picks out the
> > molecules which have an activity cliff? It selects the ones with high loss and
> > then forms edges
> > based on the relative labels - but do you know how much of the
> > time
> > these pairs do form AC pairs?
> >
> > There might be some misunderstanding.
> > This work is not on predicting whether two molecules have activity cliff. Instead,
> > we  define
> > activity cliff as a matched molecule pair
> > with different labels with respect to a given property
> > (Definition 3.2).
> > We use this definition to find molecules (in the training set)
> > with activity cliff.
> > This also follows existing works on activity cliff (such as "Exposing the limitations of molecular
> > machine learning with activity cliffs. Journal of Chemical Information and
> > Modeling", 2022).
> >
> > > **Q3.** Figure text sizes - the text on many of the figures is unreadable at a normal zoom level, try adjusting the matplotlib params to make these more readable. You can always move some into the appendix and leave one or two examples from each figure in the main text. Specially Fig 2, 3 and 6
> >
> > Thank you for your suggestion.
> > In this revised version,
> > we have made the text
> > in Figures 5 and 6
> > larger.
> > For Figures 2 and 3, we found that using a larger text size makes the figure less
> > readable. As such,
> > we provide an enlarged version of these figures as Figures 8 and 9 in Appendix C.
> >
> > > **Q4.** Caption of Fig 7 is a bit unwieldy
> >
> > Thank you for your suggestion. We now include a space between the two subfigures to make them clearly separated.
> >
> > > **Q5.** Line 194 has a typo “even they” -> “even though they” ?
> >
> > Thank you for pointing this out. We have fixed this typo.

---

> ### Author Response · Authors · 2024-12-03
>
> Dear reviewer YTtN,
>
> Thank you again for your valuable comments. As the discussion period is approaching its deadline, could you kindly take a moment to check our responses and let us know if we have adequately addressed your previous concerns? We would greatly appreciate any further feedback or comments you may have, and we are committed to addressing any outstanding issues that may have arisen.
>
> Best,
>
> Authors

---

### Official Review · Reviewer_B7eX · 2024-11-03

**Soundness:** 2
**Presentation:** 3
**Contribution:** 2
**Rating:** 3
**Confidence:** 4

**Summary:**

This paper focused on the molecular property prediction task, in particular accounting for properties exhibiting activity cliffs, which are defined as minor structural changes conferring significant changes in activity.  A new method based on curriculum learning is proposed, where property prediction is formulated as a node classification problem on a graph where nodes are molecules and edges encode molecular similarity. The proposed approach is evaluated using several classification and regression datasets and different molecular encoders.

**Strengths:**

- Except for a few points, the method is clearly described and the paper is relatively easy to follow.
- Figures help understand the intuition for the method and the method itself.
- The initial analyses are well-conducted and help familiarize with the challenge tackled in this paper.

**Weaknesses:**

- The main claims of this work appear to be not well supported by results. In particular
    - Line 70: "We are the first to investigate why...". I could not see results indicating why molecular property prediction models struggle in these casese. Instead, the work include some empirical evidence that only reinforces the (known) observation that generalizing to AC is challenging.
     - Additionally, many works investigated activity cliff in the context of property prediction. See for example "Zhang et al., Activity Cliff Prediction: Dataset and Benchmark, 2023", or "Wu et al., A Semi-Supervised Molecular Learning Framework for Activity Cliff Estimation, 2024". These works are not cited or compared.
    - Line 75: "We propose to re-formulate molecular property prediction as a node classification problem.". This does not appear completely novel, see for example "Zhuang et al., Graph Sampling-based Meta-Learning for Molecular Property Prediction, 2023" or "Zhao et al., Molecular Property Prediction Based on Graph Structure Learning, 2023", which are not cited. In general, previous works on this direction are not accounted for.
- Novelty. The novelty of the work appears limited. As stated in line 324, the methodological novelty is the extension from node to node+edge curriculum learning. However, the definition of the edge-level loss (Eq. 2) is based on the same node-level loss. This work largely appears as an application of standard curriculum learning.
- Lack of baselines. The authors only compare the proposed method to the baseline network, i.e., no other methods of any kind are taken into account. The authors should compare the proposed method to existing methods. These include works focused on AC prediction (see points above for some examples), but also methods broadly focused on representation robustness (e.g., based on adversarial perturbations or mixup) and domain generalization.

**Questions:**

See weaknesses. In particular, The authors should 1) better clarify the claims, 2) reference previous work focused on AC prediction 3) better clarify the novelty of the proposed approach, 4) include more baselines.

---

> ### Author Response · Authors · 2024-12-01
> **Responses to reviewer B7eX**
>
> We would like to thank you for your valuable suggestions. Here we reply to your comments point by point:
>
> > **W1-1.**  Line 70: "We are the first to investigate why...". I could not see results indicating why molecular property prediction models struggle in these cases. Instead, the work include some empirical evidence that only reinforces the (known) observation that generalizing to AC is challenging.
>
> Existing works focus on showing that
> accurate predictions on molecules with AC is difficult. However,
> this work goes further and investigate why making such predictions is difficult.
> As demonstrated in Figures 2 and 3,
> molecules with AC
> are harder to train,
> and they make up a large proportion of large-loss molecules and have relatively larger loss.
> Therefore, we propose to design a training algorithm to learn from these AC molecules more effectively.
>
> > **W1-2.** Additionally, many works investigated activity cliff in the context of property prediction. See for example "Zhang et al., Activity Cliff Prediction: Dataset and Benchmark, 2023", or "Wu et al., A Semi-Supervised Molecular Learning Framework for Activity Cliff Estimation, 2024". These works are not cited or compared.
>
> The suggested works are on activity cliff
> prediction, which is different
> from our task of
> molecular property prediction,
> as we do not aim to predict if two molecules have
> activity cliff.
> This is also discussed in the last paragraph of section 2.1.
> Please also refer to
> our response to Q1 in common question part
> for more discussion on the difference between our work and existing works on AC prediction.
>
> > **W1-3.** Line 75: "We propose to re-formulate molecular property prediction as a node classification problem.". This does not appear completely novel, see for example "Zhuang et al., Graph Sampling-based Meta-Learning for Molecular Property Prediction, 2023" or "Zhao et al., Molecular Property Prediction Based on Graph Structure Learning, 2023", which are not cited. In general, previous works on this direction are not accounted for.
>
> In this revised version,
> we added more discussions in Section 4.1 (highlighted in blue)
> to emphasize the differences between these works and the graph formulation in the
> proposed method LAC.
> Specifcially,
> Zhuang et al.
> uses a heterogeneous graph with different types of nodes (molecule nodes and property
> nodes)
> and edges connecting molecule nodes to the corresponding property nodes.
> Different types of edges indicate different property labels (0/1) on its connected molecule.
> However, it
> does not consider structural similarity between molecules.
> On  the other hand, in the graph of
> Zhao et al.,
> two nodes (molecules) are
> connected if they have similar embeddings computed from a pre-trained model.
> However, it does not encode
> their molecular properties (labels).
> Our graph considers both structural similarity and molecular properties of molecules.
> Two nodes (molecules) are
> connected if they form a matched molecule pair,
> and we use two types of edges to indicate whether the connected molecules have the same
> or different property labels.
> This can then better reflect the AC information.
>
> > **W2.** Novelty. The novelty of the work appears limited.
> As stated in line 324, the methodological novelty is the extension from node to node+edge curriculum learning. However, the definition of the edge-level loss (Eq. 2) is based on the same node-level loss.
> This work largely appears as an application of standard curriculum learning.
>
> There might be some misunderstanding.
> First,
> the proposed edge-level loss is NOT based from the node-level loss.
> In the experiment, we use
> the cross-entropy loss for classification problems and mean squared loss for
> regression problems.
> However,
> the edge-level loss is based on equation (2), which measures the prediction difference between two molecules with activity cliff,
> and is the same for both classification and regression problems.
>
> Second,
> the novelties of this paper include
> (i) use a new graph formulation (please also refer to our response to
> your **W1-3** above);
> (ii)
> to guide curriculum learning,
> define a new node-level weighted loss
> which is directly based on AC;
> (iii) define a new edge-level task which encourages molecules with AC to have
> different representations and property predictions.
> Hence,
> the proposed LAC is NOT a direct application of standard curriculum learning.

---

> > ### Author Response · Authors · 2024-12-01
> > **Responses to reviewer B7eX (cont.)**
> >
> > > **W3.** Lack of baselines. The authors only compare the proposed method to the baseline network, i.e., no other methods of any kind are taken into account. The authors should compare the proposed method to existing methods. These include works focused on AC prediction (see points above for some examples), but also methods broadly focused on representation robustness (e.g., based on adversarial perturbations or mixup) and domain generalization.
> >
> > As explained above, the suggested works are on AC prediction but not on molecular
> > property prediction.
> > Please also refer to our response to your **W1-2** as well as our response to **Q1 in common question part**
> > for a more detailed discussion.
> > Regarding methods on representation robustness, they are orthogonal to this paper and can be directly combined with the proposed method LAC.

---

> ### Author Response · Authors · 2024-12-03
>
> Dear reviewer B7eX,
>
> Thank you again for your valuable comments. As the discussion period is approaching its deadline, could you kindly take a moment to check our responses and let us know if we have adequately addressed your previous concerns? We would greatly appreciate any further feedback or comments you may have, and we are committed to addressing any outstanding issues that may have arisen.
>
> Best,
>
> Authors

---

### Official Review · Reviewer_gRFn · 2024-11-08

**Soundness:** 2
**Presentation:** 3
**Contribution:** 3
**Rating:** 5
**Confidence:** 4

**Summary:**

The paper proposes to use curriculum training to address the activity cliff (AC) problem in molecular property prediction. The proposed method re-formulates property prediction task as node classification where the molecules are considered as nodes, and the edges are constructed based on whether the molecules are AC pairs. The edge-level task also helps curriculum training. Extensive experiments are conducted on MoleculeNet dataset by adding the proposed method to various baselines to evaluate the model performance.

**Strengths:**

- The paper tackles a very important task in molecular property prediction, and is from the training perspective.
- The adaptation of curriculum training here is rational.
- The paper has performed a series of analyses to show the findings.

**Weaknesses:**

- Activity cliff is a critical concept in chemistry, primarily based on differences in molecular structure. Any method addressing this task should be motivated by chemical intuition related to these structural variations somehow. However, the proposed method lacks insights that would specifically address activity cliffs. It is more like tackling a hard sample problem rather than focusing explicitly on the activity cliff challenge.

- How is the "training loss for the top 10%-loss molecules with and without AC" calculated? Are the molecules involved in the matched pairs removed from this calculation? If the training involves only AC vs. non-AC molecules, the results seem fairly intuitive. Additional clarifications would help make these experiments easier to understand.

- In Fig. 4, are the dashed and solid edges used to denote different labels or categories, like being used differntly? Or are they simply shown to illustrate how AC pairs vary?

- The selection of molecules is based on loss differences, but this approach does not necessarily correlate with AC performance. For instance, an AC might result in high loss, but a high loss does not necessarily indicate an activity cliff.

- Each task seems to require a separate graph, as molecules can behave differently depending on the properties being evaluated. This approach might incur significant computational costs. Table 9 shows the results of AC pairs obtained in each dataset, which helps clarify the data size. However, the number of pairs seems quite large, making the computational complexity and scalability other concerns.

- The backbone models seem not very new. There are many recent SOTA models and the authors should evaluate the performance of adding the proposed component to these models.

- For some datasets, the method shows only marginal improvement. Have the authors explored the underlying reasons for this?

- The experiments seem only run once, which might not robust. The authors could try cross validation or run several times to report the standard deviation.

- There are already proposed AC datasets [1, 2], why the authors do not evaluate the proposed method on them?

[1] Van Tilborg, Derek, Alisa Alenicheva, and Francesca Grisoni. "Exposing the limitations of molecular machine learning with activity cliffs." Journal of chemical information and modeling 62, no. 23 (2022): 5938-5951.
[2] Zhang, Ziqiao, Bangyi Zhao, Ailin Xie, Yatao Bian, and Shuigeng Zhou. "Activity cliff prediction: Dataset and benchmark." arXiv preprint arXiv:2302.07541 (2023).

**Questions:**

See weaknesses.

---

> ### Author Response · Authors · 2024-12-01
> **Responses to reviewer gRFn**
>
> We would like to first thank you for your recoginition of our work as well as your valuable suggestions. Here we reply to your comments point by point:
>
> > **W1.** the proposed method lacks insights that would specifically address activity cliffs. It is more like tackling a hard sample problem rather than focusing explicitly on the activity cliff challenge.
>
> The proposed method does not only focus on hard samples but also considers if the sample has activity cliff. From equation (1), setting $p=1$ corresponds to selection based only on the training loss but not on activity cliff. However, as can be seen from Table 5, $p=1$ leads to worse performance than the proposed LAC.
>
> > **W2.** How is the "training loss for the top 10\%-loss molecules with and without AC" calculated? Are the molecules involved in the matched pairs removed from this calculation? If the training involves only AC vs. non-AC molecules, the results seem fairly intuitive. Additional clarifications would help make these experiments easier to understand.
>
> There might be some misunderstanding.  For the experiments in Section 3, the model is trained on the whole data set (with both AC and non-AC molecules). Figure 3 examines the training progress for the large-loss molecules (top-10\% of training loss). These large-loss molecules are split into the two groups of AC and non-AC. We can see that for these large-loss molecules, those with AC still exhibit
> larger training loss than those do not have AC. We have also added more explanations in Section 3 (highlighted in blue).
>
> > **W3.** In Fig. 4, are the dashed and solid edges used to denote different labels or categories, like being used differntly? Or are they simply shown to illustrate how AC pairs vary?
>
> In the figure, molecules are connected when they have similar structures (as defined in Definition 3.1). A solid line indicates that these two molecules have the same label (property), while a dashed line indicates that they have different labels.
>
> > **W4.** The selection of molecules is based on loss differences, but this approach does not necessarily correlate with AC performance. For instance, an AC might result in high loss, but a high loss does not necessarily indicate an activity cliff.
>
> There might be some misunderstanding.  In Section 4.2, curriculum learning is guided by the weighted loss, which is defined as $\hat{\ell}_i(w) =p_i \ell_i(w)$ with $p_i$ defined in (1). Setting $p_i=1$ corresponds to selection based only on the training loss but not on activity cliff.  However, the proposed method uses $p_i<1$, which has better performance than not using AC (i.e., $p_i=1$) as shown in Table 5.
>
> > **W5.** Each task seems to require a separate graph, as molecules can behave differently depending on the properties being evaluated. This approach might incur significant computational costs. Table 9 shows the results of AC pairs obtained in each dataset, which helps clarify the data size. However, the number of pairs seems quite large, making the computational complexity and scalability other concerns.
>
> Using separate graphs does not incur significant computational costs for data sets with multiple properties (tasks). We can first go through all molecules to find all matched molecule pairs (using Definition 3.1), then obtain the graphs for all properties by simply comparing their multiple labels in a single round. Please also refer to our response to **Q2 in common question part** for more discussion on the computational cost of our proposed method.
>
> > **W6.** The backbone models seem not very new. There are many recent SOTA models and the authors should evaluate the performance of adding the proposed component to these models.
>
> The backbone models are selected from the SOTA models on the data sets used in the experiments (sections 5.1 and 5.2). We are not aware of other models that outperforms the selected models on these data sets. Moreover, the proposed method can indeed be directly combined with other models.
>
> > **W7.** For some datasets, the method shows only marginal improvement. Have the authors explored the underlying reasons for this?
>
> There is marginal improvement only on a few combinations of data sets and models (such as 3D-PGT and UniMol models on the ClinTox data set). Since ClinTox only contains a small number of AC pairs, it is reasonable that LAC is not particularly effective. We have added some discussions in Section 5.1 (highlighted in blue) on factors that may affect the performance of the proposed method.

---

> ### Author Response · Authors · 2024-12-01
> **Responses to reviewer gRFn (cont.)**
>
> > **W8.** The experiments seem only run once, which might not robust. The authors could try cross validation or run several times to report the standard deviation.
>
> Following most works in molecular property prediction (such as 3D Informax [a] and 3D-PGT [b]),
> we report the mean over three runs.  In this revised version,
> we added results on standard deviation in Table 10 (of Appendix C.1).
>
> [a] 3D Infomax improves GNNs for Molecular Property Prediction. ICML 2022
>
> [b] Automated 3D Pre-Training for Molecular Property Prediction. KDD 2023
>
> > **W9.** There are already proposed AC datasets [1, 2], why the authors do not evaluate the proposed method on them?
>
> Our experiments on regression data sets indeed follow [1], and is now highlighted in
> blue in Section 5.2 of this revised version.
> We have also discussed about [2] in
> section 2.1
> (highlighted in blue), which considers a task different from ours.
> The task in [2] predicts whether a given pair of molecules have activity cliff,
> while our task is to predict the molecule property.
> Please also refer to
> our response to **Q1 in common question part** for more discussion on the difference between our work and existing works on AC prediction.

---

> ### Author Response · Authors · 2024-12-03
>
> Dear reviewer gRFn,
>
> Thank you again for your valuable comments. As the discussion period is approaching its deadline, could you kindly take a moment to check our responses and let us know if we have adequately addressed your previous concerns? We would greatly appreciate any further feedback or comments you may have, and we are committed to addressing any outstanding issues that may have arisen.
>
> Best,
>
> Authors

---

### Author Response · Authors · 2024-12-01
**Responses to some common questions**

We would like to thank all reviewers for checking our work and make many valuable suggestions. Here we reply to some questions commonly mentioned in the reviews:

> **Q1.** Relation to works on AC prediction/estimation.

While there are several works on AC prediction/estimation [1,2],
they focus on a different task and require a different model.
Specifically, they consider predicting whether a pair of molecules have activity cliff.
Their input is a pair of structurally similar molecules, and their output is a
binary label
(1 indicates that these two molecules may have activity cliff and their properties can be very different,
while 0 indicates they may not have activity cliff and their
properties are similar).
On the contrary, our method LAC does not aim to predict whether the test molecules have activity cliff.
Instead, we leverage the activity cliff information in the training set to
help training a molecular property prediction model.
By incorporating this information, we enable the model to learn from molecules with activity cliff more effectively,
As demonstrated
in Tables 2 and 3 and the loss visualizations in Figures 5 and 6,
this leads to improved performance.

[1] Activity Cliff Prediction: Dataset and Benchmark. arXiv preprint 2302.07541.

[2] A Semi-Supervised Molecular Learning Framework for Activity Cliff Estimation. IJCAI 2024

> **Q2.** Additional computational cost of LAC.

The computational cost of our method LAC depends on whether
the dataset with activity cliff pairs is available.
When this dataset is available (e.g., as is provided in
(van Tilborg et al., 2022)), the graph can be constructed easily and
the proposed method
(Algorithm 1)
has almost the same time cost as
standard random-sampling training,
as is demonstrated in the new
Table 13 (in Appendix C.3 of the updated version).
However,
when this data set is not available, currently we use a brute-force approach to examine all molecule pairs, which takes time quadratic in the number of molecules (as can also be seen from Table 13).
More effective algorithms to find activity cliff pairs from a given molecular
property prediction dataset is beyond the scope of this work,
and will be addressed
in the future.

---

### Meta-Review · Area_Chair_UWp7 · 2024-12-18

**Metareview:**

**Summary:** The authors propose a molecular property prediction that takes into account the activity cliff (AC) between two molecules. Specifically, the authors design a graph where the nodes correspond to molecules and the edges indicate whether the two molecules have an activity cliff. They validate their method on a variety of tasks.

**Strengths:** The paper is well-written and the method is clearly explained. The authors successfully demonstrate the challenges with property prediction that stem from AC via an empirical study. The performance of the method is good and it is easy to be incorporated into existing pipelines.

**Weaknesses:** Some reviewers raise concerns about the novelty and lack of baselines. The authors claim that the previous works that have been pointed out by the reviewers are specifically on AC prediction.

**Decision:** From my understanding, the authors aim to solve a different problem than AC prediction, i.e., predicting the properties of the molecules, and AC is only used to encapsulate more information about the relation between the molecules. Constructing the graph by first predicting whether or not the two molecules have AC is a separate problem and is not the focus of the current paper. Given the positive feedback by two reviewers, I believe the proposed method is useful and shows promise for using AC to improve molecular property prediction. Thus, I lean toward acceptance.

**Additional Comments On Reviewer Discussion:**

The authors clarified some of the misunderstandings and tried to address the concerns raised by the reviewers. Reviewer B7eX is still not convinced with the response, but I believe the authors tried their best to respond to their questions.

---

### Decision · Program_Chairs · 2025-01-22

Accept (Poster)